# Influence of P3HT:PCBM Ratio on Thermal and Transport Properties of Bulk Heterojunction Solar Cells

**DOI:** 10.3390/ma16020617

**Published:** 2023-01-09

**Authors:** Dorota Korte, Egon Pavlica, Domen Klančar, Gvido Bratina, Michal Pawlak, Ewa Gondek, Peng Song, Junyan Liu, Beata Derkowska-Zielinska

**Affiliations:** 1Laboratory for Environmental and Life Sciences, University of Nova Gorica, Vipavska 13, SI-5000 Nova Gorica, Slovenia; 2Laboratory for Organic Matter Physics, University of Nova Gorica, Vipavska 13, SI-5000 Nova Gorica, Slovenia; 3Institute of Physics, Faculty of Physics, Astronomy and Informatics, Nicolaus Copernicus University, 87-100 Torun, Poland; 4Institute of Physics, Cracow University of Technology, 30-084 Kraków, Poland; 5State Key Laboratory of Robotics and System, Harbin Institute of Technology, Harbin 150001, China; 6School of Instrumentation Science and Engineering, Harbin Institute of Technology, Harbin 150001, China; 7School of Mechatronics Engineering, Harbin Institute of Technology, Harbin 150001, China

**Keywords:** organic bulk-heterojunction solar cells, polymer-fullerene solar cells, photothermal beam deflection spectrometry, non-radiative recombination, thermal diffusivity, frequency domain methods

## Abstract

The influence of P3HT:PCBM ratio on thermal and transport properties of solar cells were determined by photothermal beam deflection spectrometry, which is advantageous tool for non-destructively study of bulk heterojunction layers of organic solar cells. P3HT:PCBM layers of different P3HT:PCBM ratios were deposited on top of PEDOT:PSS/ITO layers which were included in organic bulk-heterojunction solar cells. The thermal diffusivity, energy gap and charge carrier lifetime were measured at different illumination conditions and with a different P3HT:PCBM ratios. As expected, it was found that the energy band gap depends on the P3HT:PCBM ratio. Thermal diffusivity is decreasing, while charge carrier lifetime is increasing with PCBM concentration. Energy band gap was found to be independent on illumination intensity, while thermal diffusivity was increasing and carrier lifetime was decreasing with illumination intensity. The carrier lifetime exhibits qualitatively similar dependence on the PCBM concentration when compared to the open-circuit voltage of operating solar cells under AM1.5 illumination. BDS and standard I-V measurement yielded comparable results arguing that the former is suitable for characterization of organic solar cells.

## 1. Introduction

Semiconducting polymers are becoming increasingly important due to their potential application in novel organic optoelectronic devices such as organic solar cells (OSCs), organic field-effect transistors (OFET) or organic light-emitting diodes (OLED) [1,2]. As for OSCs, substantial effort has been invested recently to improve their power conversion efficiency (PCE). The approaches include chemical [3], structural modification [4,5] and thermal treatment [3,6] of the layers, which form OSCs.

It is becoming increasingly present that PCE exhibits a complex dependence between charge carrier extraction, charge carrier mobility and recombination rate of photogenerated charge carriers [7,8]. The charge carrier extraction depends on the length of the pathway, along which they are photogenerated, and the extraction efficiency depends on the recombination process, which can be either radiative or non-radiative [9]. Moreover, recombination significantly depends on the density of charge carriers, which can be either photogenerated or injected from the electrodes [10,11].

Quantitative determination of recombination coefficient is a complex task due to its time dependence and dependence on the chemical composition of the layers forming OSCs.

Chemical composition of the layers within OSCs, in turn affects not only their charge carrier transport properties but also their thermal properties. Therefore, understanding the processes that govern heat transport along these layers, could in principle, help in optimization of the chemical and morphological properties of the layers, with the aim of maximizing PCE [12]. In order to study the thermal transport through the layers of OSCs under operational conditions, a non-contact and non-invasive experimental method is required. Available experimental techniques that satisfy these requirements include a steady-state infrared thermography, laser flash method, thermo-reflectance, modulated photothermal radiometry and photoacoustic method. Among these the frequency-domain methods are preferred [13] because they allow one to perform depth profilometry, yield two simultaneous information on the amplitude and phase, and provide lock-in detection for low-signal conditions. For characterization of the OSCs a depth profilometry is of particular interest since it allows one to analyze of multilayered samples. By changing of the modulation frequency of the excitation light, the thermal diffusion length is changed. Therefore, it is possible toa discriminate the signals emitted from the topmost layers of the sample (10 nm–100 μm, several 10–100 kHz modulation frequencies) from the signals emitted from the deeper layers and the substrate. Next, providing the two simultaneous channels of information increases the level of confidence in the experimental data. The lock-in detection ensures a superior signal-to-noise ratio comparing to the time-domain or steady state methods. Common to all optical methods is a high spatial resolution due to the fact that excitation laser beam can be focused onto a submicron spot size.

In this work we focused on the characterization of thermal properties and charge carrier transport in OSC comprising blended layers of poly(3-hexylthiophene) (P3HT) and [14,14]-phenyl-C_61_-butyric acid methyl ester (PCBM) using beam deflection spectrometry (BDS). Both P3HT and PCBM are well established materials used to fabricate bulk-heterojunction OSCs in which both materials are blended in a single active layer. Electronic energy band gap of P3HT is ~1.9 eV, with the majority of absorbed light from the visible part of the solar spectrum [15,16]. The bulk heterojunction layer was prepared on top of the hole-transport layer of poly(3,4-ethylenedioxythiophene) polymer, crosslinked with the poly(styrenesulfonic acid) (PEDOT:PSS) [17]. PEDOT:PSS layer is characterized by high hole conductivity, optically transparent and forms suitable interface dipole with P3HT:PCBM for photogenerated charge extraction [14]. Under the PEDOT:PSS there was an indium-tin-oxide (ITO) cathode and on top of the P3HT:PCBM there was an aluminum anode.

BDS is based on the illumination of the sample by light of selected wavelength, intensity and time duration. The analysis of the absorbed light allows one to extract information on the chemical composition and optical properties of the sample. The absorbed light is in part converted to heat within the sample and by measuring it is possible to determine thermal diffusivity and thermal conductivity of the sample. Both parameters, in turn, can be used to describe, at least in a qualitative fashion, mechanical, transport, structural properties of the sample. Due to direct dependence of a BDS signal on the intensity of excitation light, these techniques offer higher sensitivity compared to conventional reflectance and transmission techniques, which have recently been further improved by optimization of the pump/probe beam geometries and by performing the measurements in organic solvents as contact fluids [18].

One of the advantages of BDS for characterization of OSCs is also the possibility to perform non-contact and non-destructive analysis of the multilayered structures [19] without the need of electrical contacts that are required in the case, e.g., photoconductivity measurements. Furthermore, BDS technique is extremely sensitive to the absorption and subsequent de-excitation processes of the investigated material. Both these processes are determined by the energy band gap, carrier lifetime and thermal properties of the layer. BDS is therefore suitable for in-situ measurements, allowing continuous monitoring of the processes governing the synthesis of the layers, and determination of thermal, optical and transport properties of multilayered materials [20,21,22,23]. The use of BDS to characterize the charge and thermal transport offers the possibility to determine thermal diffusivity and conductivity, energy band gap and carrier lifetime of organic solar cells in a single analysis, while performing the measurements in a non-contact and non-destructive way [24,25,26]. Knowledge of them is crucial to determine its applicability in the field of OSCs since they define the rate of light absorption, generation of electron-hole pairs or excitons, as well as separation of charge carriers. These processes further effect the efficiency of light conversion into electricity.

The advantage of BDS is demonstrated by a set of measurements performed on organic solar cell under illumination. In order to directly and independently measure transport and recombination parameters, the BDS measurements were correlated to the current-voltage characterization of organic solar cell under standard AM1.5 illumination.

## 2. Materials and Methods

### 2.1. Preparation of Solar Cells

OSC (see Figure 1) were prepared on an indium tin oxide (ITO) coated glass slides, which were purchased from Sigma-Aldrich (Darmstadt, Germany) (surface resistivity 8–12 Ω/cm^2^). A 2 mm wide region of ITO was removed from the edges of the substrates by etching in 37% hydrochloric acid for 20 min. Consequently, the substrates were thoroughly cleaned in a series of ultrasound baths of acetone, isopropanol, 2% Hellmanex solution and deionized water. A co PEDOT:PSS purchased also from Sigma-Aldrich was spin-coated at 7000 rpm for 60 s and maximum acceleration. Under these conditions, the thickness of the PEDOT:PSS layer amounts to approximately 70 nm. After PEDOT:PSS deposition, the substrate was heated to 120 °C for 10 min on a hot plate and transferred into the glove box an inert 99.999% N_2_ atmosphere, with oxygen level below 10 ppm and water level below 5 ppm.

A poly(3-hexylthiophene-2,5-diyl) polythiophene (P3HT) polymer was blended with [14,14]-phenyl-C61-butyric acid methyl ester (PCBM) fullerene. The blend was prepared as a 1:1, 1:3 and 3:1 weight ratio in 1,2-dichlorobenzene solution of 5 mg/mL P3HT and 15 mg/mL PCBM. The solution was mixed overnight at room temperature. In order to remove any non-dissolved material, the solution was filtered through a Whatman 0.2 µm PTFE membrane syringe filter.

Active layers were prepared by spin coating the blend solution at 1000 rpm for 60 s with the acceleration of 1000 rpm/s. The resulting layer thickness was about 100 nm as presented in Figure 1. Solution preparation and layer deposition was performed in a nitrogen glove box. Upon completion of the layer deposition the samples were transferred to a vapor deposition system mounted inside the same glove box. Here, 100 nm thick Al topmost electrode was deposited by vacuum evaporation through a shadow mask. The area of each device was defined by the overlap of the ITO layer and the top Al electrode. This area was measured by a micrometer and was approximately 50 mm^2^. Solar cells were heated to 130 °C for 6 min on a hot-plate inside the same glovebox. Thermal annealing temperature and duration was not optimized in terms of device photovoltaic performance.

### 2.2. Current-Voltage Characteristics of Solar Cells

Current-voltage (I-V) characteristics of OSCs were measured in the nitrogen atmosphere in the same glove box, in which the cells were prepared. The electric contacts to solar cell electrodes were realized by pressing small indium pieces on the ITO and the Al layer. This method resulted in a stable, low resistance pads, which were contacted with golden probes. Solar cell characteristics were calculated from the I-V characteristics measured with a Keithley, Cleveland, OH, USA, 2400 SourceMeter in dark and under illumination by a class AM1.5 standard reference spectrum solar simulator with an intensity of 100 mV/cm^2^ (Model SS-50AAA, Photo Emission Tech, Ventura, CA, USA).

### 2.3. Photothermal Beam Deflection Spectrometry (BDS)

The schematic diagram of the BDS experimental arrangement is shown in Figure 2.

A continuous He-Ne laser (wavelength λ = 633 nm, MELLES GRIOT, Bensheim, Germany, Model 25-LHP-928-230) with output power of 35 mW (spot size of 2 mm, intensity of 70·10^−6^ Wm^−2^) was applied as incident heating beam (excitation beam). It was modulated by a broadband electro-optical (EO) amplitude modulator (New Focus, Santa Clara, CA, USA, Model 4102-M) for a wavelengths and frequency range 600–900 nm and DC-200GHz, respectively, driven by a high voltage amplifier (New Focus, Model 3211) and followed by a polarizer (Thorlabs, Bergkirchen, Germany). A He-Ne laser (wavelength λ = 543.5 nm, MELLES GRIOT, Bensheim, Germany, Model 25-LGR-393-230) was used as probe beam source of the 2 mW (spot size of 50 μm, intensity of 10^−7^ Wm^−2^) output power. The probe beam intensity change, after its interaction with the temperature oscillation (TOs), was detected by four-quadrant position sensitive detector (RBM—R. Braumann GmbH, Attenkirchen, Germany, Model C30846E) and processed by means of a lock-in amplifier (Stanford research instruments, Sunnyvale, CA, USA, Model SR830 DSP). An x-y-z-translation stage (CVI, Albuquerque, NM, USA, Model 2480M and 2488) allowed to vary the position in x, y and z direction to optimize the experimental configuration. The sample was illuminated by a Krypton laser (λ = 407 nm, Innova 300C, Coherent, Lisses, France), which was used to generate carriers in the examined materials and 60 and 120 mW power at the location, where TOs were induced.

In order to prevent deterioration of organic semiconductor during photothermal characterization, the solar cells were sealed in an inert nitrogen atmosphere as schematically shown in Figure 1. Theoretically, the geometry was approximated as a 6-layered structure surrounded with an infinite column of air. The heat source represents the heat produced by the absorption of the excitation light beam (EB). Therefore, it was assumed that EB is only absorbed in the central layer comprising of P3HT:PCBM and PEDOT:PSS layers. Other layers exhibit negligible light absorption.

## 3. Results

The absorbance at 633 nm and 407 nm of a 100 nm thick P3HT:PCBM layer depends on the P3HT and PCBM ratio and varies from the value of 0.3 to 0.4 [27,28,29]. It means that most of the excitation beam energy is absorbed by the P3HT:PCBM layer. The remaining part of the energy is absorbed by the underlying PEDOT:PSS layer. Hence, it is expected that the BDS signal (see Appendix A) contains information mostly about thermal, optical and transport properties of the P3HT:PCBM layer. In order to determine these properties, the BDS amplitude and phase dependence on the modulation frequency of EB were measured in the frequency range from 0.1 Hz to 5 Hz. The chosen modulation frequency range was related to the thermal diffusion length *μ*th in the nitrogen, glass and air layers above P3HT:PCBM, in which the TOs were generated. We found that *μ*th is between 8.4–1.2 mm, 14.2–2.1 mm and 8.4–1.2 mm for nitrogen, glass and air, respectively, in a selected frequency range of 0.1–5 Hz (according to Appendix A). The obtained results are presented in Figure 3, Figure 4, Figure 5 and Figure 6 for three different P3HT:PCBM ratios and three different illumination modes. In can be seen that in all cases the amplitude rapidly drops with increase in modulation frequency since the induced TOs are strongly attenuated in the medium they propagate and the damping coefficient increases with increase in *f*. Further increase in the modulation frequency results in slower decay of the amplitude value. In contrast, phase exhibits more monotonous decay with the increase in modulation frequency of EB. Comparing the amplitude as the function of the P3HT:PCBM ratio, the highest amplitude of 5.0 μV is obtained in case of sample with 3:1 P3HT:PCBM ratio, that is followed by 1:1 and finally 1:3 (the amplitude of 3 μV). Red circles correspond to the BDS signal resulting from the fundamental sample illumination by the 633 nm excitation beam of 35 mW, whereas green triangles represent the BDS signal resulting from illumination of the sample by the fundamental excitation beam (633 nm) in addition to the 407 nm illumination of 60 mW output power. The blue squares correspond to the BDS signal resulting from sample illumination of the 633 nm together with illumination of the 407 nm and 120 mW output power. It can be seen that the amplitude and phase of measured BDS signal (Figure 3, Figure 4 and Figure 5) increase with the increase in the output power of the light illuminating the sample. In fact, the amplitude of BDS signal is expected to increase with the illumination power due to increase in the induced density of heat sources (see in Appendix A) that are a consequence of non-radiative de-excitation processes (realized in a form of heat) of the absorbed energy from EB.

In order to investigate the thermal and transport properties of the examined blends, the theoretical model (described by Appendix A) was developed to extract the desired properties from the experimental data by the use of multiparameter fitting procedure based on the least-squares method (solid line in Figure 3, Figure 4 and Figure 5). The validity of the method was tested by simulating data, performing the fitting procedure to that data and checking if the received results agree with those used for the simulation. The fitted parameters were the effective sample thermal diffusivity (*D*), energy band gap (*E*_g_), carrier life time (*τ*), surface recombination velocity (*v_SR_*) and its thickness (*l*_2_), whereas the fixed parameters where those of experimental setup as the sample and detector position, probe beam radius and its waist position, power of pump beam, thermal properties of the fluid in which the TOs are induced (thermal diffusivity and conductivity) as well as thermal and structural properties of all other layers constructing the whole sample (Figure 1) (thermal diffusivities and conductivities, thicknesses of both glass and ITO layers). The change in the thermal diffusivity is followed by the change in charge carrier lifetime as result of the change in carrier concentration. That are estimated by the use of equation *n* = (2*γτ*)^−1^, where *τ* is the carrier life time, *γ* is the recombination coefficient that is of the order of 10^−13^ cm^3^s^−1^ [30]. The effective absorption coefficients were found by the use of a least-square method fitting procedure. The obtained values were: 2·10^4^ cm^−1^, 6·10^4^ cm^−1^, 1·10^5^ cm^−1^ for 1:3 P3HT:PCBM, 1:1 P3HT:PCBM and 3:1 P3HT:PCBM, respectively.

The obtained parameter values are presented in Table 1, Table 2 and Table 3. The P3HT:PCBM thickness is equal for all examined samples. In contrast, *E*_g rr_ increases with the concentration of the PCBM in the blend from 1.75 ± 0.10 eV for the ratio of 3:1 to 2.25 ± 0.10 eV for the ration 1:3. This behavior is presumably the result of the phenomena, which was previously observed in semiconducting polymer blended with small photochromic molecule [31]. Once blended, the photochromic molecules preserved their electronic structure in parallel to the electronic structure of the polymer matrix. Similarly, P3HT:PCBM blends exhibit a combination of electronic levels of individual component. Thus, *E*_g_ in organic solar cells reflects the energy difference between HOMO of the donor component and LUMO of the acceptor component [10]. In our case, HOMO of P3HT as a donor is approximately−5 eV, while LUMO of PCBM acceptor is around −3 eV [32]. Therefore, the expected difference is 1.4 eV, which is slightly lower than our results. However, the experimental method presented here provides a more precise measurement of effective *E*_g_, which apparently increases with the PCBM concentration. This information can be advantageously used to study other phenomena related to the electronic structure of organic semiconducting blends.

In contrast to *E*_g_, thermal diffusivity (*D*) and carrier lifetime (*τ*) strongly depend on the excitation light power [33]. *D* has increased by about 30% when illuminating the sample with the additional 60 mW laser and has further increased by another 30% by increasing its power twice (Table 1, Table 2 and Table 3). Hence *D* is not directly proportional to the light power, but it is to the thermal conductivity normalized to mass density and specific heat capacity (see Appendix A). Mass density and heat capacity are not expected to be significantly changed with the power of excitation light, although the temperature of the examined layers raise due to light excitation. The observed *D* change presumably reflects the change in material thermal conductivity. Thermal conductivity of organic semiconductors is determined by phonons and electrons [1]. The population of both heat carriers is generally low in the lack of light illumination. For instance, Hiura et al. measured *D* of P3HT film at the value of ~10^−7^ m^2^/s [34]. In contrast, once illuminated, the absorbed photons excite excitons—coupled pair of electron and hole. Excitons in organic semiconductors are coupled to local perturbation of the electronic cloud, distortion of structure of the molecule and distortion of the surrounding molecules or polymer segments [35]. This phenomenon of coupling between a single or a pair of charge carriers and “dressing” phonons can be described as a virtual particle known as a polaron. Therefore, the photogenerated heat carriers of organic semiconductors are polarons. Thus, the change in *D* with the illumination intensity depends on the density of photogenerated polarons in P3HT:PCBM layers that are relaxed non-radiatively by converting the absorbed energy to a heat producing the BDS signal. Although the absolute value of *D* differs among layers of different content of PCBM, the 60% increase between the lowest and the highest illumination intensity demonstrate the role of photogenerated polarons in the heat transport of organic solar cells.

The lowest value of *D* is obtained for P3HT:PCBM ratio of 1:3 whereas the highest value of *D* is obtained for P3HT:PCBM ratio of 3:1. These results indicate that *D* is raising with the P3HT fraction. In fact, thermal conductivity and diffusivity of PCBM were found to be significantly reduced due to strong localization of vibrations localized on fullerene derivatives due to molecular side chains [36]. We therefore concluded that the photogenerated polarons, which end on PCBM moieties, are lost in terms of heat transport. Consequently, *D* is dropping with the increase in PCBM content in the whole layer. Furthermore, the transport of photogenerated polarons depends on the structure and chemical composition on the nanoscale. This is proved by measurements of thermal conductivity of P3HT:PCBM blends as well as PEDOT:PSS in Ref. [37], which agree with our results presented here and indicate that the thermal conductivity *k* of blends depends on the ratio between individual components. In case of thermally annealed blends changes in the values of *k* were observed. It should be stated that thermal annealing of organic solar cells is used to remove residual solvents and to induce partial phase separation to form nanostructured bulk heterojunction structure [38,39]. The nanostructured P3HT:PCBM layer consists of a combination of nanometer-sized crystalline segments of polymer and fullerenes, surrounded by amorphous polymer phases. It turns out that the thermal transport is highly impeded at the boundaries between these segments [1,40,41]. In fact, the thermal transport is enhanced along nanocrystals, nanorods and nanowires [34,42]. Therefore, thermal conductivity follows similar dependence on the morphology than the charge carrier conductivity [43], which can be tuned by the modification of nanostructure of organic semiconductors [44,45].

In the case of BDS measurement, the thermal diffusivity is the product of the mobility of photogenerated polarons and their density, which increases nonlinearly with light intensity. Photogenerated charge carriers results from the dissociation of photogenerated exciton and subsequent separation of electron-hole pairs [46].

The free charge generation in P3HT:PCBM materials is very efficient [47,48], which means that there are almost no losses due to inefficient exciton decay. However, the currents in the presented solar cells are fairly small, which may indicate other loss processes.

Charge recombination of photogenerated polarons can be divided into radiative and non-radiative recombination. As shown in Table 1, Table 2 and Table 3 *τ* decreases with the illumination intensity. The recombination in organic solar cells can be approximated with the bimolecular Langevin recombination [49], which increases with the square of the charge carrier density. Hence, the decrease of *τ* results in the increase in carrier concentration and, consequently in the surface recombination velocity that was found to be of about 10^3^ and 10^4^ cm·s^−1^ for the carrier concentrations of 10^22^ and 10^23^ m^−3^, respectively.

In parallel to BDS measurements, we have studied P3HT:PCBM solar cell under AM1.5 illumination. I–V characteristics of OSCs under illumination and in dark are presented in Figure 6, and typical figures of merit of parameters are summarized in Table 4. Serial and parallel resistance were extracted as described in [50,51].

It can be seen that the highest and the lowest concentration of PCBM results in the best values of power efficiency (1.30% and 1.78%, respectively) of the three. We note that all other figures of merit of the OSCs presented here are lower than the present standard values. This is because our experiments were not targeted to optimize OSCs performance.

Figure 7 represents *τ* dependence as a function of P3HT:PCBM ratio. The shortest lifetime is observed in case of 3:1 P3HT:PCBM ratio, and the longest for 1:3 P3HT:PCBM ratio. This behavior is opposite to the behavior of thermal diffusivity. Knowing that *τ* is limited by the non-radiative recombination, since only that one produces BDS signal, the 3:1 ratio exhibits the highest non-radiative recombination. The recombination was found to reduce the open-circuit voltage of organic solar cells and is discussed in the forthcoming paper [19]. The negative correlation between the thermal diffusion length and VOC, which is related to exciton lifetime results from the fact that thermal diffusion length *μ*th determines the depth in the material that is penetrated by induced TOs. Longer *μ*th, implies higher concentration of the photoexcited charge carriers. Higher charge density results in a higher recombination rate, and consequently shorter lifetime.

The light intensity impinging onto the samples of the AM1.5 source during I-V characterization was 0.1 W cm^−2^. On the other hand, the power of incident excitation light during BDS experiments was of 35 mW, 95 mW and 155 mW, focused to the spot diameter of 2 mm. This resulted in the light intensity of 1.1 W cm^−2^, 3.0 W cm^−2^and 4.9 W cm^−2^ for 35 mW, 95 mW and 155 mW, respectively. Although AM1.5 illumination comprises of full spectrum of the visible light, the excitation during BDS is approximately an order of magnitude higher. Nevertheless, we observe similar trend of the *V_oc_* compared to *τ* (Figure 7). *V_oc_* in an ideal, recombination-free OSC would retain the value of *E_g_*. In real OSCs*V_oc_* is reduced by the radiative and non-radiative recombination [52]. Since *τ* is proportional to the recombination rate, V_oc_ is proportional to ln(*τ*^−1^). This is demonstrated in Figure 7, where *V_oc_* is presented by a boxplot as a function of P3HT:PCBM ratio.

## 4. Conclusions

In this work, a novel approach of determining the thermal and charge transport properties of organic solar cells based on photothermal effect was presented. From the measurement point of view, it should be emphasized that using the frequency domain method, such as photothermal beam deflection method (BDS), it is possible to investigate a multi-layered structure especially when the layer under investigation exhibits the highest optical absorption. Moreover, the presented method is a unique, sensitive and universal spectroscopic technique, which allows one to determine the thermal and transport properties during a single non-destructive measurement in different environmental conditions. To demonstrate the capabilities of BSD we characterized thermal diffusivity, energy gap and charge carrier lifetime of prototypical organic solar cell comprising of P3HT:PCBM layers on top of PEDOT:PSS and ITO. Using different illumination intensities, we found that thermal diffusivity exhibits a strong dependence on the density of the photogenerated polarons and on the ratio between P3HT and PCBM fraction. The measured energy gap was dependent on the P3HT:PCBM ratio and the photogenerated carrier lifetime was decaying inversely to thermal diffusivity. The latter was assigned to the non-radiative recombination and compared to electrical properties of organic solar cells under AM1.5 illumination. Similar dependence exhibited charge carrier lifetime on the open-circuit voltage of illuminated solar cells. We also found that the best photovoltaic properties were found for P3HT:PCBM in the ratio of 1:3 and 3:1. The obtained parameters indicate that the studied solar cells show attractive photovoltaic behavior and can be modified due to the different P3HT: PCBM ratio.

## Figures and Tables

**Figure 1 materials-16-00617-f001:**
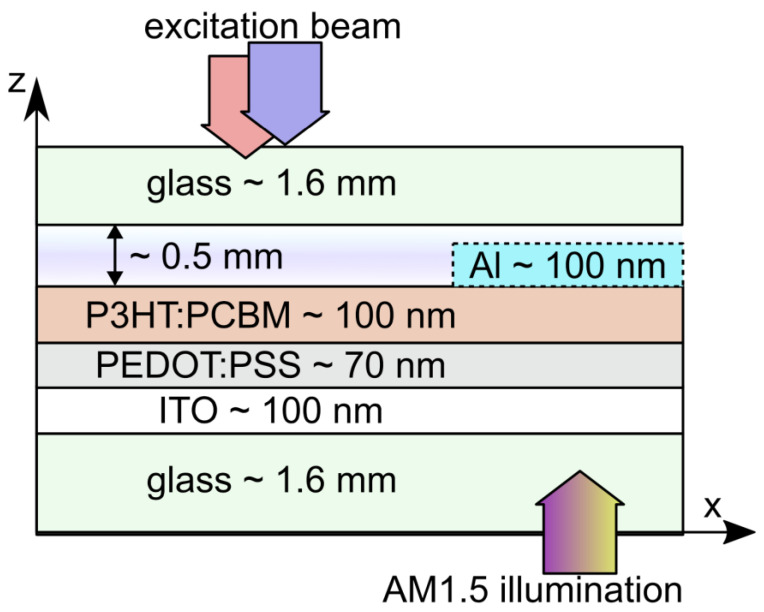
Schematic of an organic solar cell, which was assembled in an inert nitrogen atmosphere maintained between two sealed glass plates. The top Al electrode (dashed lines) was not present during photothermal spectroscopy. Light-absorbing layer comprises of a P3HT:PCBM layer and a PEDOT:PSS layer. These layers were illuminated from the top during photothermal spectrometry and from the bottom during electrical characterization of the solar cell as indicated by arrows.

**Figure 2 materials-16-00617-f002:**
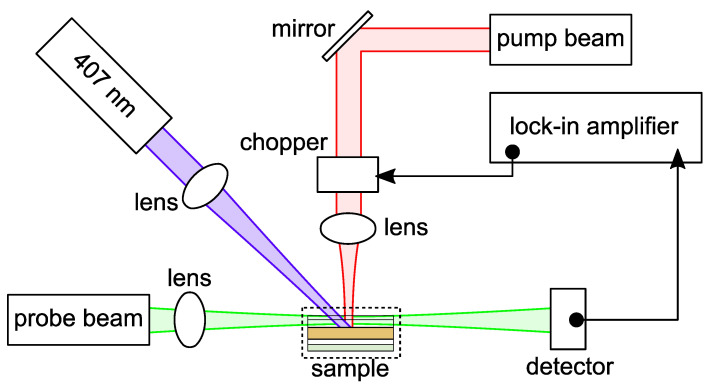
Schematic diagram of the experimental setup of the BDS method. Probe beam intensity is 2 mW. Pump beam intensity is 35 mW. An additional 407 nm laser was used to continuously photoexcite the active layer of organic solar cell, labeled as sample.

**Figure 3 materials-16-00617-f003:**
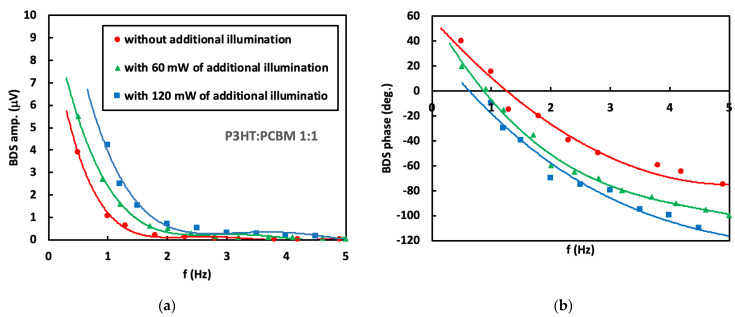
(**a**) Amplitude and (**b**) phase of BDS signal dependence on modulation frequency of TOs for P3HT:PCBM (1:1).

**Figure 4 materials-16-00617-f004:**
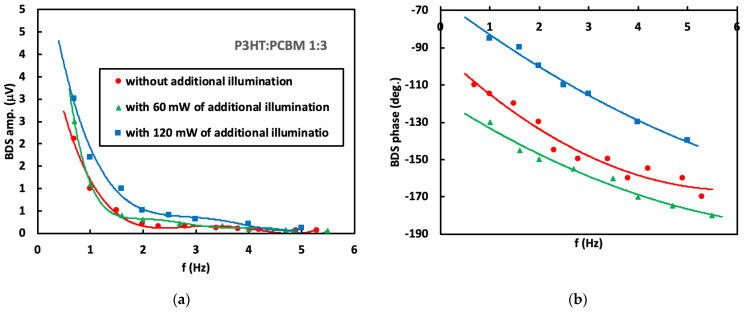
(**a**) Amplitude and (**b**) phase of BDS signal dependence on modulation frequency of TOs for P3HT:PCBM (1:3).

**Figure 5 materials-16-00617-f005:**
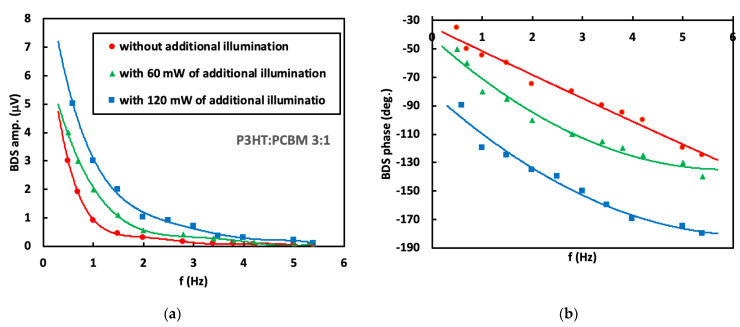
(**a**) Amplitude and (**b**) phase of BDS signal dependence on modulation frequency of TOs for P3HT:PCBM (3:1).

**Figure 6 materials-16-00617-f006:**
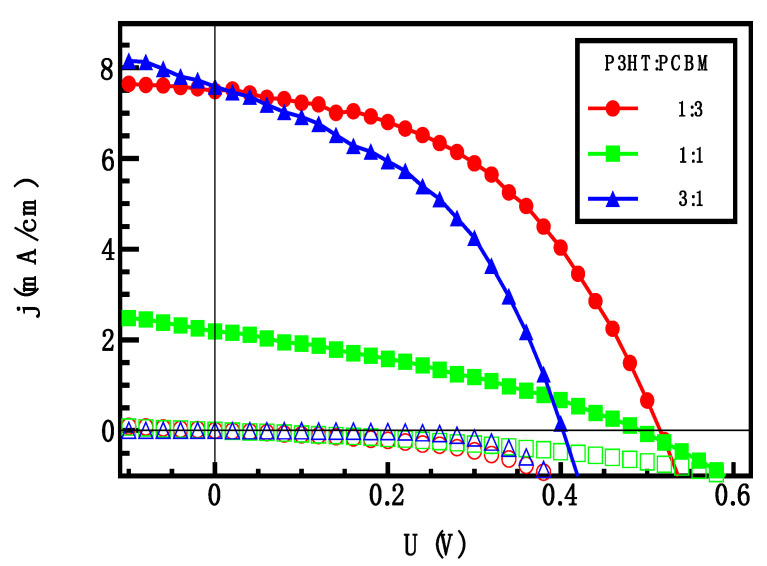
Current-voltage characteristics of solar cells in dark (hollow symbols) and under AM1.5 illumination (filled symbols) as a function of P3HT:PCBM ratio.

**Figure 7 materials-16-00617-f007:**
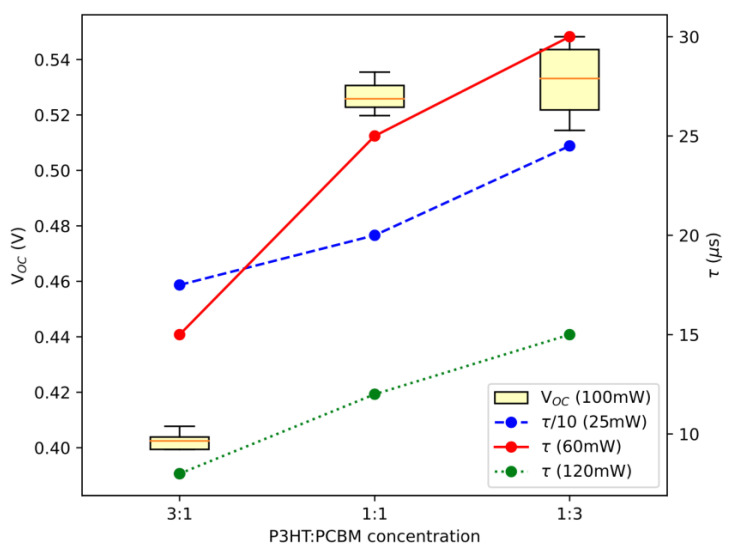
Boxplot of V_OC_ as a function of P3HT:PCBM ratio, compared to charge carrier lifetime (lines) at different illumination intensity.

**Table 1 materials-16-00617-t001:** Effective parameters of P3HT:PCBM(1:1)+PEDOT:PSS determined by BDS.

λ [nm]	P [mW]	*D* [m^2^s^−1^]	*E_g_* [eV]	*τ* [μs]	Thickness [nm]	n [m^−3^]
630	35	(0.158 ± 0.012)·10^−6^	2.10 ± 0.10	200 ± 15	175 ± 20	(2.50 ± 0.19)·10^22^
400	60	(0.202 ± 0.014)·10^−6^		25 ± 2		(2.00 ± 0.16)·10^23^
400	120	(0.268 ± 0.018)·10^−6^		12 ± 1		(4.17 ± 0.35)·10^23^

**Table 2 materials-16-00617-t002:** Effective parameters of P3HT:PCBM(1:3)+PEDOT:PSS determined by BDS.

λ [nm]	P [mW]	*D* [m^2^s^−1^]	*E_g_* [eV]	*τ* [μs]	Thickness [nm]	n [m^−3^]
630	35	(0.075 ± 0.005)·10^−6^	2.25 ± 0.10	245 ± 15	175 ± 20	(2.04 ± 0.13)·10^22^
400	60	(0.112 ± 0.009)·10^−6^		30 ± 5		(1.67 ± 0.28)·10^23^
400	120	(0.148 ± 0.011)·10^−6^		15 ± 1		(3.33 ± 0.22)·10^23^

**Table 3 materials-16-00617-t003:** Effective parameters of P3HT:PCBM(3:1)+PEDOT:PSS determined by BDS.

λ [nm]	P [mW]	*D* [m^2^s^−1^]	*E_g_* [eV]	*τ* [μs]	Thickness [nm]	n [m^−3^]
630	35	(0.183 ± 0.11)·10^−6^	1.75 ± 0.10	175 ± 15	175 ± 20	(2.86 ± 0.25)·10^22^
400	60	(0.241 ± 0.018)·10^−6^		15 ± 2		(3.33 ± 0.44)·10^23^
400	120	(0.325 ± 0.023)·10^−6^		8 ± 1		(6.25 ± 0.78)·10^23^

**Table 4 materials-16-00617-t004:** Principal OSCs figures of merit: JSC—Current Density, VOC—Open circuit Voltage, FF—fill factor, η—power efficiency.

Photovoltaic Cell	JSC [mA/cm^2^]	VOC [V]	FF	η [%]
P3HT:PCBM (3:1)	7.55	0.41	0.42	1.30
P3HT:PCBM (1:1)	2.20	0.50	0.31	0.34
P3HT:PCBM (1:3)	7.47	0.52	0.46	1.78

## Data Availability

Not applicable. Data supporting the results of this study are available from the appropriate author upon reasonable request.

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
