# Peer review of "Influence of P3HT:PCBM Ratio on Thermal and Transport Properties of Bulk Heterojunction Solar Cells"

_materials, 2023, doi:10.3390/ma16020617_

Round 1

Reviewer 1 Report

A novel approach, photothermal beam deflection method (BDS), to determine the thermal and charge transport properties of organic solar cells based on photothermal effect was presented.

Anyway, some points of BDS could be more emphasized and showed, for example, how important and what actually potential of BDS compared to other techniques and the present solar cell materials.

Please re-check the explanation of Fig.3 in the content (absence), as well as re-structure of Fig.3 in the content. The sentences ahould not be presented in between the figure.

Author Response

Thank you very much for your comments.

Please find below our answers.

Point 1: Anyway, some points of BDS could be more emphasized and showed, for example, how important and what actually potential of BDS compared to other techniques and the present solar cell materials.

Response 1: The proper description was added to the article text to Introduction section as “The experimental technique epresented in this work belongs to the group of frequency-domain methods. It is known as photothermal beam deflection spectrometry (BDS). It is based on the absorption of the excitation radiation, what enables chemical analysis and optical characterization of the sample, whereas the consequent conversion of absorbed energy into heath during radiationlessdeexcitation process enable the determination of thermal parameters (thermal diffusivity and conductivity) and related properties (mechanical, transport, structural etc) of the examined sample. Due to direct dependence of BDS signal on the intensity of excitation light, these techniques offer higher sensitivity compared to conventional reflectance and transmission techniques, which have recently been further improved by optimization of the pump/probe beam geometries and by performing the measurements in organic solvents as contact fluids [20].

The advantage of BDS for characterization of OSCs is also its suitability for non-contact and non-destructive analysis of the multilayered structures [21] without the necessity of deposition   electrical contacts that are required in case of other characterization methods, e.g. photoconductivity measurements. Furthermore, BDS technique is directly sensitive to the absorption and subsequent de-excitation processes of the investigated material, which in turn are determined by its energy band gap, carrier lifetime and thermal properties. Thus, it is applicable for in situ measurements allowing continuous monitoring and determination of thermal, optical and transport properties of different multilayered materials [22-25] in a simple and inexpensive way. Furthermore, the complex structure of the examined materials requires determination of numerous parameters for their proper characterisation. This in turn necessitates larger amounts of samples, and application of several instrumental techniques, which makes characterization of the material very much time consuming. It is therefore advantageous to use novel emmerging techniques, such as BDS, for their characterization by determination of multiple parameters. The use of BDS to characterize the charge and thermal transport offers the possibility to determine thermal diffusivity and conductivity, energy band gap and carrier life time of organic solar cells in a single analysis, while performing the measurements in a non-contact and non-destructive way. This enables preservation of samples, which can be further used in investigations, e.g. in studies of electrical properties [26-28].”

Point 2: Please re-check the explanation of Fig.3 in the content (absence), as well as re-structure of Fig.3 in the content. The sentences ahould not be presented in between the figure.

Response 2: The descritpiopn is added to the manuscript text as: “The obtained results are presented in Figures 3-5 for three different P3HTCBM ratios and three different illumination modes. In can be seen that in all cases the amplitude rapidly drops with increase in modulation frequency since the induced TOs are strongly attenuated in the medium they propagate and the damping coefficient increases with increase in f. Further increase in the modulation frequency results in slower decay of the amplitude value. In contrast, phase exhibits more monotonous decay with the increase in modulation frequency of EB. Comparing the amplitude as the function of the P3HTCBM ratio, the highest amplitude of 5.0 mV is obtained in case of sample with 3:1 P3HTCBM ratio, that is followed by 1:1 and finally 1:3 (the amplitude of 3 mV). Open triangles correspond to the BDS signal resulting from the fundamental sample illumination by the 633 nm excitation beam of 35 mW, whereas open circles represent the BDS signal resulting from illumination of the sample by the fundamental excitation beam (633 nm) in addition to the 407 nm illumination of 60 mW output power. The open squares correspond to the BDS signal resulting from sample illumination of the 633 nm together with illumination of the 407 nm and 120 mW output power. It can be seen that the amplitude and phase of measured BDS signal (see Figs. 3-5) increases with the increase in the output power of the light illuminating the sample. In fact, the amplitude of BDS signal is expected to raise with the illumination power due to increase in the induced density of heat sources (see in Eq.(SM.3) in Supplementary Materials) that are the consequences of non-radiative de-excitation processes (realized in a form of heat) of the absorbed energy from EB.”

The structure of Fig. 3 is also corrected.

Reviewer 2 Report

1. Check the unit of intensity carefully.

2. Mention the value of the ratio of PEDOT:PSS material which you have used in your experiment.

3. Why did you choose only  three values of ratio of P3HT:PCBM ? Why not 2:1 and 1:2 etc.? Why did not perform its optimization experiment? Clarification is required.

4. What was the annealing temperature and duration of heating ?

5. Rewrite the whole RESULTS section and resubmit. 

Author Response

Thank you very much for your comments. Please find below our answer for them.

Point 1:  Check the unit of intensity carefully.

Response 1: The units were checked and the manuscript was properly corrected.

Point 2:   Mention the value of the ratio of PEDOT:PSS material which you have used in your experiment.

Response 2: PEDOT:PSS material was purchased from Sigma-Aldrich (code: 560596-25G) and used as obtained. From the specifications list we understand that it is a 2.8 wt.% dispersion of PEDOT:PSS in water. However, we have no information about the ratio. The only available information is the resistivity, which is between 10-300 kOhm-cm and viscosity, which is below 20cP.

Point 3:   Why did you choose only  three values of ratio of P3HT:PCBM ? Why not 2:1 and 1:2 etc.? Why did not perform its optimization experiment? Clarification is required.

Response 3: The message of our work is to demonstrate the advantages of photothermal beam spectrometry to characterize solar cells. Therefore we prepared solar cells with three different ratios. Since the message of the work is not to demonstrate that the ratio of blend influence the solar cell efficiency, we did not perform the optimization of the ratio of P3HT:PCBM in terms of solar cell efficiency.

Point 4:    What was the annealing temperature and duration of heating ?

Response 4: Thermal annealing temperature was 130ºC for 6 minutes. Accordingly, we have modified the corresponding sentence in the section 2.1 and moved it to the end of the section: "Solar cells were heated to 130ºC for 6 minutes on a hot-plate inside the same glovebox. Thermal annealing temperature and duration was not optimized in terms of device photovoltaic performance."

Point 5:    Rewrite the whole RESULTS section and resubmit. 

Response 5: The section was rewritten.

Reviewer 3 Report

Dear Editor,

I have read the manuscript carefully. The study can be useful for the reader of the journal. I recommend the publication of the manuscript after the corrections given below.

1- Although the structure was sealed in the N2 atmosphere, it is not possible that the structure has a N2 Layer. Please erase it to misunderstanding 

2- There are some problems with figure captions in the text. Please check them. (Like "Figures Error! Reference source not found.-Figure 4":)  

3- the references are old. Please update them

Author Response

Thank you very much for your comments. Please find below our answer for them.

Point 1:    Although the structure was sealed in the N2 atmosphere, it is not possible that the structure has a N2 Layer. Please erase it to misunderstanding 

Response 1: Thanks for this suggestion. We have removed the label N2 and fixed the size of the top-glass to avoid misunderstanding.

Point 2:     There are some problems with figure captions in the text. Please check them. (Like "Figures Error! Reference source not found.-Figure 4":)  

Response 2: The caption was checked and corrected.

Point 3:    the references are old. Please update them

Response 3: The references were updated but those who contain very basic information were kept.

Reviewer 4 Report

This work must be improved significantly. The current form can not be accepted because of the following reasons:

1) Introduction is so long, and it is difficult to understand the target, problem, and solution clearly in the authors' work.

2) Experiments: There are only two measurements including I-V and BDS. It is simple and not of enough quality for an investigation.

3) The results are poor. There are no other characteristics to support the conclusions clearly. There are no deep discussions about mechanisms and characteristics.

4) Presentation is not good. It is difficult to follow the figures and discussions. The quality of the figures was so low.

Author Response

Thank you very much for your comments. Please find below our answer for them.

Point 1:    Introduction is so long, and it is difficult to understand the target, problem, and solution clearly in the authors' work.

Response 1: The Introduction section was rewritten.

Point 2:    Experiments: There are only two measurements including I-V and BDS. It is simple and not of enough quality for an investigation.

Response 2: The I-V measurements were performed on the selected solar cell devices. The selected devices, although not the best in terms of power conversion efficiency, were simultaneosly characterized using standard I-V and novel BDS method. In fact, the message of the publication is to demonstrate the agreement among the two characterization methods. Based on the presented results, we can clearly see the correlation between two different characterization methods. Nevertheless, we agree with the referee's comment that the statistics can be further improved with more measurements.  

Point 3:     The results are poor. There are no other characteristics to support the conclusions clearly. There are no deep discussions about mechanisms and characteristics.

Response 3: We are confused by the referee's comment. The discussion about the observed trends of Voc and lifetime is presented in lines 380-406. In the given discussion we relate our finding with the studies, which were presented by other groups. We found that the results of other studies agree with our findings. Here, we would like to stress again that the message of our work is to compare standard I-V characterization with BDS method. For that purpose, we did not include any more thorough discussion about the recombination, which we find to be the most important mechanism for the observed behavior of Voc and lifetime.     

Point 4:     Presentation is not good. It is difficult to follow the figures and discussions. The quality of the figures was so low.

Response 4: The figures were corrected.

Round 2

Reviewer 2 Report

It is suggested to mentioned 

1) Fig. 3(a and b) instead of Fig.3

2) Fig. 4(a and b) instead of Fig.4

3) Fig. 5(a and b) instead of Fig. 5

4) If possible can be made better English.

Except these suggestions, this manuscript is suitable for publication.

Author Response

Point 1: It is suggested to mentioned

1) Fig. 3(a and b) instead of Fig.3

2) Fig. 4(a and b) instead of Fig.4

3) Fig. 5(a and b) instead of Fig. 5

4) If possible can be made better English.

Except these suggestions, this manuscript is suitable for publication.

Response 1: All reviewer's comments were corrected. Accordingly, the Figures 3, 4 and 5 were labeled with (a) and (b). In addition the text was thoroughly checked for typos and English was improved.

Reviewer 4 Report

Although the Introduction part has been improved to show clearly the target, the experiment section, and the results are still limited.  In addition, there are still errors in the Figure presentation (Figure 5) and English writing in the revised version. I feel that the quality of the revised manuscript is still not enough for publication. I suggest that the authors should improve strongly the manuscript and resubmit it.

1. Main focus of this work is the influence of P3HT: PCBM ratio on thermal and transport properties..., however, only three weight ratios of P3HT: PCBM (1:1, 1:3, and 3:1) were considered. Are they typical ratios? Or randomly chosen ratios? In my opinion, only the three ratios considered are not quantitive enough to show typical results. 

2. BDS method is not new. It is used commonly to examine film quality. Therefore, the idea in this work is not novel. It should be combined with other methods to demonstrate the thermal and transport properties. 

3. The I-V results showed the PCE of devices was so low. It demonstrates that the quality of the film was not so good with the ratios. Therefore, the conclusions from this work are difficult to convince the readers.

Author Response

Thank you for your comments, which improved the quality of our work. We did our best to improve the English.

Point 1: Main focus of this work is the influence of P3HT: PCBM ratio on thermal and transport properties..., however, only three weight ratios of P3HT: PCBM (1:1, 1:3, and 3:1) were considered. Are they typical ratios? Or randomly chosen ratios? In my opinion, only the three ratios considered are not quantitive enough to show typical results.

Response 1: We believe that the reviewer was confused about what is the main message of our work. The message of our work is to demonstrate the advantages of photothermal beam spectrometry to characterize solar cells. Therefore we prepared solar cells with three different ratios. Since the main  message of the work is not to demonstrate that the ratio of blend influences the solar cell efficiency, we did not perform the optimization of the ratio of P3HT:PCBM in terms of solar cell efficiency. In order to avoid similar conclusions of readers, we modified a sentence in the abstract from:

"It was found that the energy band gap depends on the P3HT:PCBM ratio."

to

"As expected, it was found that the energy band gap depends on the P3HT:PCBM ratio."

Point 2: BDS method is not new. It is used commonly to examine film quality. Therefore, the idea in this work is not novel. It should be combined with other methods to demonstrate the thermal and transport properties.

Response 2: We agree that BDS method is not new. However, the results of our work demonstrate that BDS can be used to access the semiconducting properties of complex heterojunction materials and structures, which are present in organic solar cells. In other words the novelity of the work is the application of BDS for such analysis. For that purpose it was necessary to developed a new theoretical model to extract the desired properties from the BDS measurements in one single measurement without the necessity of using other methods. The obtained results coincide with literature data.

Point 3: The I-V results showed the PCE of devices was so low. It demonstrates that the quality of the film was not so good with the ratios. Therefore, the conclusions from this work are difficult to convince the readers.

Response 3: As previously answered in comment 1, we believe that the reviewer drawn wrong conclusion about the main message of the manuscript. The I-V measurements were performed on the selected solar cell devices. The selected devices, although not the best in terms of power conversion efficiency, were simultaneously characterized using standard I-V and BDS method. Since the message of the publication is to demonstrate the agreement among the two characterization methods, devices were not optimized in terms of PCE. In order to avoid similar judgement of readers, we modified the abstract as presented in the answer to comment 1.